# Somatostatinoma and Neurofibromatosis Type 1-A Case Report and Review of the Literature

**DOI:** 10.3390/diagnostics10090620

**Published:** 2020-08-21

**Authors:** Sorina Martin, Simona Fica, Ovidiu Parfeni, Liliana Popa, Teodora Manuc, Oana Rizea, Ioana Lupescu, Mirela Gherghe, Gabriel Becheanu, Adina Croitoru

**Affiliations:** 1Endocrinology Department, Carol Davila University of Medicine and Pharmacy, 020021 Bucharest, Romania; simonafica55@gmail.com; 2Endocrinology Department, Elias Hospital, 011461 Bucharest, Romania; parfeniovidiu@gmail.com; 3Dermatology Department, Carol Davila University of Medicine and Pharmacy, 020021 Bucharest, Romania; lilidiaconu@yahoo.com; 4Dermatology Department, Elias Hospital, 011461 Bucharest, Romania; 5Gastroenterology Department, Fundeni Clinical Institute, 022328 Bucharest, Romania; teodora.manuc@gmail.com; 6Radiology Department, Carol Davila University of Medicine and Pharmacy, 020021 Bucharest, Romania; rizea.oana@gmail.com (O.R.); ioana.lupescu@umfcd.ro (I.L.); 7Radiology Department, Fundeni Clinical Institute, 022328 Bucharest, Romania; 8Nuclear Medicine Department, Alexandru Trestioreanu Oncological Institute, 022328 Bucharest, Romania; 9Pathology Department, Carol Davila University of Medicine and Pharmacy, 020021 Bucharest, Romania; gbecheanu@yahoo.com; 10Laboratory of Histopathology and Immunohistochemistry, Victor Babes National Institute of Pathology, 050096 Bucharest, Romania; 11Oncology Department, Titu Maiorescu University of Medicine, 040051 Bucharest, Romania; adina.croitoru09@yahoo.com; 12Oncology Department, Fundeni Clinical Institute, 022328 Bucharest, Romania

**Keywords:** neuroendocrine tumor, somatostatinoma, neurofibromatosis type 1, somatostatin analogues, chemotherapy, capecitabine, temozolomide

## Abstract

Somatostatinomas are rare neuroendocrine tumors (NET) that arise in the gastrointestinal (GI) tract. Because of their insidious growth, they are usually asymptomatic until late stages, presenting as malignant disease. We report the case of a 50-year-old woman who presented with epigastric abdominal pain, diarrhea and significant weight loss in the last two years. On clinical examination the patient met the criteria for neurofibromatosis type 1 (NF1). Abdominal CT and MRI revealed an infiltrative duodenal mass, with pancreatic invasion, locoregional enlarged lymph nodes and disseminated hepatic nodules. Microscopy and immunohistochemistry uncovered a neuroendocrine tumor, staining positive for chromogranin A (CgA), synaptophysin and somatostatin, with a Ki67 = 1%. Somatostatin receptors (SSTRs) type 2 were negative and SSTRs type 5 were positive in less than 50% of tumoral cells. Our patient was classified as a T3N1M1 stage IV metastatic duodenal grade 1 somatostatinoma and treatment with somatostatin analogues and chemotherapy with capecitabine and temozolomide was started, with so far abdominal imaging follow-up showing stable disease. When a patient is diagnosed with a rare NET, such as a somatostatinoma, it is of utmost importance to determine if it is a sporadic tumor or just a feature of a genetic disorder.

## 1. Introduction

Somatostatinomas are neuroendocrine tumors of D-cell origin that contain and sometimes secrete somatostatine [1]. The estimated incidence of this tumor is about one in 40 million [2] and it represents about 4% of gastrointestinal neuroendocrine tumors (GI-NETs) [3]. Approximately 55% of somatostatinomas are in the pancreas and the reminder arise in the ampulla and periampullary region of the duodenum and rarely in the jejunum [4]. Usually, somatostatinomas present as a metastatic disease, 75% of them being malignant tumors [5]. Somatostatinomas can occur in association with multiple endocrine neoplasia (MEN)-1, neurofibromatosis type 1 (NF1), tuberous sclerosis and von-Hipple Lindau syndrome (VHL), and a new syndrome, via hypoxia-inducible factor 2 somatic mutation, which includes somatostatinoma, paraganglioma/pheochromocytoma and polycythemia, found only in females, had been also described [6].

NF1 is the most common hereditary multitumor syndrome with an incidence at birth of approximately 1:3000 and it is caused by mutations in the NF1 gene. NF1 is localized on chromosome band 17q11.2 and it acts as a tumor suppressor gene. Biallelic NF1 inactivation results in complete loss of functional neurofibromin activity. Neurofibromin acts as a negative regulator of the RAS-MAPK pathway, and it has a GTPase activating domain that interacts with RAS and converts active RAS-GTP to its inactive form. Apart from NF1 promotor hypermethylation, other somatic inactivation events occur in tumors associated with NF1, such as intragenic mutations and loss of heterozygosity [7]. Persons with NF1 develop both benign and malignant tumors at increased frequency throughout life, neurofibromas being the most common type of benign tumor. Optic pathway gliomas are the predominant type of intracranial neoplasms and can lead to endocrine disorders, mostly in pediatric population [8]. The frequency of intrabdominal manifestations of NF1 ranges from 5 to 25%, and they appear during midlife, usually later than cutaneous manifestations. NF1 associates multiple gastrointestinal stromal tumors and neuroendocrine tumors of the adrenal, duodenal ampulla, pancreas and other sites [7]. Approximately 10% of the patients with NF1 develop somatostatinomas, which are typically located in the duodenum, are rarely associated with somatostatinoma syndrome and are less likely to metastasize as compared with sporadic somatostatinomas [9]. Patients presenting with this tumor are usually over 50, with a roughly equal gender distribution [2].

Somatostatin is a tetradecapeptide that inhibits numerous endocrine hormones, including insulin; pancreatic polypeptide; cholecystokinin; gastrin; glucagon; gastric inhibitory polypeptide; secretin; motilin; and exocrine functions such as gastrointestinal transit time, intestinal motility and absorption of nutrients from the small intestine. The classic somatostatinoma syndrome is described as an association of weight loss; abdominal pain; and less often diabetes mellitus, cholelithiasis and diarrhea/ steatorrhea. This syndrome occurs more often in tumors that are localized in the pancreas than those in the duodenum [9]. Most tumors appear to be well-differentiated islet cell or carcinoid-type tumors on light microscopy and they have positive somatostatin and chromogranin A (CgA) immunohistochemistry. Sporadic and NF1 related duodenal somatostatinomas frequently present with psammoma bodies [10].

Somatostatinomas are usually found during exploratory laparotomy, computed tomography (CT), magnetic resonance imaging (MRI), ultrasound or gastrointestinal endoscopy studies that are performed due to various symptoms and signs associated with somatostatinoma syndrome. The diagnosis can also be established by the presence of a fasting plasma somatostatin level exceeding 30 pg/ml [9,11]. Imaging can localize the tumor and stage the extent of disease. The evaluation of patients with somatostationoma begins with a helical, multiphasic, contrast-enhanced CT. MRI is performed when CT shows indeterminate lesions that need further characterization. Endoscopic ultrasound (EUS) can also be performed in patients with inconclusive CT or MRI. Somatostatin receptor scintigraphy or Ga-68 DOTATATE positron emission tomography (PET)/CT represent high sensitivity imaging evaluation methods and are performed if the finding of extra-abdominal metastases would change the treatment [12,13].

Usually, at diagnosis, patients present with metastatic disease, but survival rate is high, especially in those with metastatic duodenal somatostatinoma [6]. In cases of nonmetastatic somatostatinoma the surgical removal of the tumor represents the only chance of cure. Surgery can also be used in metastatic or locoregional extensive disease, for debulking, to provide symptomatic relief and prolong survival rate [14,15]. Metastatic liver disease can be treated by radiofrequency ablation, cryoablation, hepatic artery embolization or chemoembolization with doxorubicin, cisplatin and mitomycin C and selective intraarterial irradiation with yttrium labeled microspheres, alone or in combination with surgery [15,16]. Somatostatin analogues (SSAs) inhibit somatostatin secretion, providing relief of symptoms such as diarrhea, diabetes and weight loss and they are used for patients with metastatic or inoperable somatostatinoma [17,18,19]. It had been shown that lanreotide is associated with significantly prolonged progression free survival (PFS) compared to placebo [20]. Molecularly targeted therapy (everolimus, sunitinib) is used for patients with progressive advanced somatostatinomas. Conventional chemotherapy (CHT) combined with SSAs has been used as an initial treatment for highly symptomatic disease due to tumor bulk or for those who have rapidly enlarging metastases, but the experience with systemic chemotherapy in patients with somatostatinomas is limited [9]. Radiotherapy using SSAs coupled with Lutetium-177 represent a promising approach in the treatment of NETs and was associated with best tumor response, especially in patients with a high uptake on Octreoscan, limited hepatic tumor mass and a high Karnofsky performance score [20].

There are not enough data on somatostatinomas to give accurate estimates of survival, but the prognosis with any therapy is poor in patients with metastatic disease. The follow-up extends to a maximum of 10 years post-resection, and it should consist of history and physical examination, a check of the fasting plasma somatostatin level and imaging studies every 6 to 12 months or more often in the first year [9].

## 2. Case Presentation

We report the case of a 50 years old woman, without any significant personal or family history, who presented with epigastric abdominal pain, diarrhea and 30 kg weight loss in the last 2 years. The patient signed a written informed consent and the ethics committee of Elias Hospital approved the publication of this case report (no 5316/2020). The clinical examination revealed elevated blood pressure (160/100 mmHg) and the presence of a few café au lait spots located on the upper limbs; large, poorly circumscribed areas of hyperpigmentation on the trunk; bilateral axillary freckling (Figure 1 and Figure 2); and multiple asymptomatic, soft, flesh-colored, sessile or dome-shaped cutaneous tumors, 1–2 cm in diameter, nonadherent to the underlying structures, distributed on the trunk and limbs, highly suggestive of neurofibromas (Figure 3). Several painless, violaceous, subcutaneous pseudoatrophic macules, with a rubbery consistency, ranging in size from 5 mm to 2 cm, typically found in NF1, were also present on the abdominal wall (Figure 4).

The mucous membranes did not show any pathologic changes and Lisch nodules were absent. Based upon the presence of three out of seven characteristic clinical features, our patient met diagnostic criteria developed by the United States National Institutes of Health (NIH) Consensus Conference in 1987 and updated in 1997 and was clinically diagnosed with neurofibromatosis type 1 (NF1) [21]. Biological evaluation was unremarkable except for diabetes mellitus (HbA1c% = 7%, fasting glycemia = 136mg/dl) and vitamin D deficiency (25 hydroxy-vitamin D = 10.53 ng/mL).

Upper digestive endoscopy revealed a gastric mucosa with prepyloric, circumferential linear and round-oval ulcerations, the largest of 6 mm, with the base covered with fibrin debris and an infiltrated area of 7 mm that could be seen between the first and second portion of the duodenum, on the medial wall, with rigidity and exaggerated friability upon biopsy. The normal papilla was visualized distally to the infiltrated area. A multiphasic contrast-enhanced CT diagnosed a 32 × 37 × 47 mm infiltrative duodenal mass, with heterogenous contrast enhancement, which narrowed the duodenal lumen and compressed the pancreatic head, and multiple low-density hepatic nodules strongly suggestive of metastatic disease. On MRI, the tumor displayed irregular borders with high contrast subtraction and heterogenous texture; it was located near the ampulla, in the second segment of the duodenum, with pancreatic invasion, associated locoregional enlarged lymph nodes and multiple disseminated hepatic nodules, with maximum diameters of 12 × 10 mm, suggestive of miliary metastatic disease (Figure 5 and Figure 6).

Pathological exam of the duodenal biopsy described in our case a tumor with glandular and cribriform structures in the *lamina propria*, with an infiltrative aspect, focal foveolar metaplasia of the villous epithelia (Figure 7), cribriform structures with cuboidal or low columnar epithelial cells, with eosinophilic cytoplasm and round, central, monotonous nuclei, without visible nucleoli (Figure 8) and the tumor had a glandular (acinar) growth pattern with calcospherites (psammoma bodies). Immunohistochemistry described a tumor positive for chromogranin A, synaptophysin and somatostatin with a Ki67 of 1% (Figure 9). Somatostatin receptors (SSTR) type 2 were negative and SSTR type 5 were positive in less than 50% of tumoral cells.

Our patient was evaluated using the radiopharmaceutical labelled with 99 mTc, available at the moment of diagnosis; 740 MBq 99mTc-EDDA/HYNIC-Tyr3-Octreotide was injected intravenously, with an acquisition protocol of early (1 h after injection) and late (4 h after injection) whole body scans with a two-headed, large field-of-view gamma camera equipped with low energy high resolution (LEHR) collimator and SPECT-CT acquisition 2 h after injection, on the region of interest (abdomen). On somatostatin receptor scintigraphy (SRS), SPECT-CT acquisition, we noticed the duodenal tumor compressing the pancreatic head and the particular aspect of very small disseminated nodules in both hepatic lobes, but without any uptake of the specific radiopharmaceutical; the aspect correlates with the immunohistochemistry results (Figure 10 and Figure 11).

In this context we performed an endocrinological screening for MEN1 and pheochromocytoma which were negative. Plasma chromogranin A and 24 h urinary 5-hydroxyindoleacetic acid (5- HIAA) were in normal ranges. Unfortunately, plasma levels of somatostatin could not be measured at diagnosis.

According to the American Joint Committee on Cancer TNM staging system for neuroendocrine tumors of the duodenum and ampulla of Vater [22], our patient was classified as a T3, N1, M1 stage IV metastatic duodenal, grade 1, SSTR negative somatostatinoma. The patient was discussed in our multidisciplinary team meeting. NCCN guidelines version 1.2019 recommend for clinically significant tumor burden: lanreotide or octreotide and/or everolimus, peptide receptor radionuclide therapy with radiolabeled somatostatin analogues (PRRT), hepatic-directed therapy or interferon alfa-2b or cytotoxic chemotherapy if no other options feasible [23]. The MRI appearance of miliary liver metastases, with significant tumor burden, led us to decide on the association of SSAs with chemotherapy, as in our country we did not have other available reimbursed therapeutic options. We began the treatment with lanreotide Autogel 120 mg every 28 days and capecitabine 750 mg/m2/BID on days 1–14, and temozolomide 200 mg/m2/ID on days 10–14 every 28 days (CAPTEM). Follow-up at 3 (Figure 12), 6, 9 and 12 months with abdominal MRI and thorax CT showed stable disease. In May 2020, follow-up CT scan at 16 months from diagnosis showed a 14 × 8 mm segment 6 liver lesion, suggestive of a new metastasis, which needed to be monitored (Figure 13).

There was a very good tolerability of both SSAs and chemotherapy. The patient presented grade 2 thrombocytopenia at cycles 3 and 4 and grade 2 palmar-plantar erythrodysesthesia at cycles 3, 4 and 5. Diarheea disappearead after the first two months of therapy.

## 3. Discussion

NF1 is an autosomal dominant disorder with complete penetrance and variable expression, associated with an increased risk of developing benign and malignant tumors of the skin, central and peripheral nervous system, adrenals and gastrointestinal tract, including pancreatic tumors [7,24]. The NF1 gene mutations determine a reduction or a complete loss of function of the cytoplasmatic protein, neurofibromin, which regulates cellular proliferation by inactivating p21 RAS and the MAP kinase [7,25]. The diagnosis of NF1 is based upon the presence of characteristic clinical features. According to NIH criteria, at least two of the following clinical features must be present to make the diagnosis of NF1: six or more café au lait macules (0.5 cm in children or >1.5 cm in adults), two or more cutaneous/subcutaneous neurofibromas or one plexiform neurofibroma, axillary or groin freckling, optic pathway glioma, two or more Lisch nodules (iris hamartomas seen on slit lamp examination) and bony dysplasia (sphenoid wing dysplasia, bowing of long bone ± pseudoarthrosis), first degree relative with NF1 [21]. As in our case, approximately 50% of NF1 patients have no family history of disease because the NF1 gene has high mutation rates in humans [26]. The multidisciplinary team management of this particular case also included genetic counseling. Genetic testing can be considered for a molecular diagnosis and can help direct screening of family members. Often, genetic testing is not required, but can it be a useful tool in confirming the diagnoses of children who do not meet diagnostic criteria or only exhibit café-au-lait macules and axillary freckling. Although of great academic interest, given the unquestionable clinical diagnosis, the fact that our patient did not have any first degree relatives that could benefit from the testing and that a positive NF1 mutation result is not a predictor of the severity or complications of the disorder, with some specific exceptions, we decided not to perform a genetic testing. The disorder is associated with an 8–15-year reduction in average life expectancy in both genders, primarily due to malignant neoplasms and cardiovascular disorders. Malignant peripheral nerve sheath tumor, breast cancer, cutaneous neurofibromas and significant psychiatric and neurologic diagnoses are common problems in patients with NF1 [27]. Patients with NF1 are also prone to developing gastrointestinal and retroperitoneal lesions, such as true neurogenic neoplasms, interstitial cell of Cajal lesions, NETs and adrenal tumors (pheochromocytoma) [7]. In 1982, Cantor et al. first documented the association between duodenal somatostatinoma and NF1 [28]. The association was afterwards described as a common gastrointestinal manifestation in patients with NF1 [11,29]. A case report and review of the literature, published in 1995, described 29 cases of duodenal somatostatinoma unassociated with NF1 and 27 cases of duodenal somatostatinoma associated with NF1, out of which four were also found to have adrenal pheochromocytomas [30]. An analysis of 162 patients with somatostatinomas (81 pancreatic somatostatinomas vs. 81 duodenal somatostatinomas) found a statistically significant difference between these two groups, regarding the incidence of NF1 (20% vs. 43.2%) [4]. In a more recent analysis, Garbrecht et al. confirmed the occurrence of duodenal somatostatinoma in NF1 patients, but with a lower frequency (14%), and found no pancreatic somatostatinoma associated with NF1 [10]. A 2010 review of literature showed that 47% of periampullary tumors in NF1 patients were of neuroendocrine origin with 40% being reported as somatostatinoma, 6% as carcinoid and 1% as malignant endocrine tumor [31].

Somatostatinoma is a rare NET [9] that originates in the enteric endocrine cells [32]. More than half of them are located in the pancreas and the rest in the ampulla or periampullary region of the duodenum or rarely in the jejunum. The tumors can be classified as functioning or non-functioning, depending on the secretion of somatostatin and often can present as metastatic disease upon diagnosis [9]. Duodenal somatostatinomas frequently display obstructive jaundice, abdominal pain, cholelithiasis, vomiting and abdominal bleeding [11] and can be accompanied by a somatostatin syndrome (steatorrhea, diabetes mellitus, cholelithiasis, weight loss), usually only if they are larger than 4 cm in size [33]. In our case, although the tumor was larger than 4 cm, located near the duodenal ampulla, with pancreatic invasion, the patient did not develop obstructive jaundice or cholelithiasis, but, instead, she presented diarrhea, weight loss and secondary diabetes mellitus. Although high plasma somatostatin levels (>30 pg/mL) may be a valuable tool to establish the biological diagnosis of somatostatinoma [9] and our patient associated somatostatin syndrome, we were not able to measure the somatostatin levels before the treatment initiation. Cg A is a neuroendocrine protein located in the secretory vesicles of neurons and endocrine cells, plays an important role in vesicle formation, is the precursor for many functional peptides (vasostatin, pancreastatin, catestatin, parastatin, etc.) and can be measured in order to provide information about diagnosis, prognosis and follow-up in NETs [34,35]. However, Cg A concentration may be normal in some low proliferative index NETs, as was the case for our patient. Pheochromocytomas are associated with VHL disease, MEN and NF1. In patients with NF1 they can occur with a frequency of 0.1–5.7%, higher than in the general population [36]. Moreover, there is also a hypoxia-inducible factor 2 somatic mutation, detected in ampullary somatostatinomas as part of a syndrome of paraganglioma and somatostatinoma, associated with polycythemia and found only in females [6]. Our patient had normal plasma normetanephrines and metanephrines, but further regular screening for other malignancies associated with NF1 is mandatory [27].

Duodenal neuroendocrine tumors account for about 20% of gastrointestinal tract neuroendocrine tumors and may show G-cell differentiation (gastrinomas) or D-cell differentiation expressing somatostatin, termed somatostatinomas, if they are functional. Acinar and psammomatous somatostatinomas are commonly found in the periampullary region in patients with NF1 and represent one third of somatostatin expressing tumors. Sometimes patients with NF1 may present other mesenchymal lesions: neurofibromas and gastrointestinal stromal tumors. Biopsy was performed and light microscopy revealed the presence of psammoma bodies, which had been previously identified in 68% of duodenal somatostatinomas [11]. Immunohistochemistry showed positive staining for Cg A, synaptophysin and somatostatin, Ki67 % = 1%, SSTR5 positive in less than 50% of tumor cells and negative SSTR2. Somatostatin receptor-based imaging can provide useful information on overall tumoral burden and location, and positive imaging can also confirm the presence of SSTRs [37]. SRS is a molecular imaging procedure for NET diagnosis and staging; it is more sensitive and specific at the biological than anatomical level, in contrast to conventional imaging, which it complements; the detection rate was reported to be between 80–100% in different studies [38,39,40]. There are two radiopharmaceuticals available on the market 111In-pentetreotide ([111In-DTPA0]-octreotide) with affinity for SSTR 2 and SSTR 5 and 99 mTc-EDDA/HYNIC-Tyr3-Octreotide, with high affinity for SSTR 2, but with a lesser extent for SSTR 3 and SSTR 5. 68Ga-DOTATATE is superior to other imaging modalities [6], but it was not available; therefore, we performed a 99 mTc-tektrotyde scintigraphy with no tumoral uptake for radiolabeled SSA.

Conventional and functional imaging revealed in our case, on diagnosis, a stage IV duodenal somatostatinoma (T3, N1, M1) with pancreatic invasion, locoregional lymph nodes enlargement and multiple disseminated hepatic nodules described as miliary metastases. Although it seems unreasonable to include SSAs in the therapeutic approach of a mostly SSTR negative somatostatinoma, in patients with symptomatic disease, clinically significant tumor burden or progressive disease, there is evidence in favor of SSA therapy (octreotide LAR or lanreotide) that can improve the related symptoms by reducing somatostatin secretion and potentially control tumor growth [20,41]. The correct mechanism by which SSAs inhibit tumoral somatostatin secretion is poorly understood. A possible explanation could be that octreotide has a higher affinity for the somatostatin receptor than for natural somatostatin and the enzymatic degradation of synthetic SSAs is on a slower rate than that of natural somatostatin [19]. High expression of SSTR 2 in NETs was associated with improved PFS and overall survival (OS) in patients treated with SSAs. Yet, there was no clear correlation between high vs. low expression of SSTRs 1–5 and uptake on somatostatin scintigraphy [42]. In one study, three out of ten patients with SSTR 2 negative and SSTR 5 positive NETs were responsive to SSA therapy [43]. Novel strategies such as SSTR 2 receptor gene transfer might prove useful as new therapies for unresectable tumors [44]. Diarrhea and weight loss, presenting signs in our patient and components of the somatostatinoma syndrome, ameliorated in the first couple of months as a result of SSA therapy. Intriguing enough, some studies have shown that patients with metastatic or inoperable NETs and no radiolabeled analogue uptake on Octreoscan or 68Ga-DOTA-peptides PET/CT, may also respond well to SSA therapy [43]. Although SSAs are highly effective at improving the hormone hypersecretion-associated symptoms, there is limited experience and little evidence that SSA therapy inhibits tumor growth, especially in somatostatinomas. Considering the results of randomized controlled studies for metastatic midgut NETs (octreotide) and gastroenteropancreatic NETs (lanreotide) without hormone-mediated symptoms, a reduction in tumor growth is expected. Therefore, due to the miliary aspect of liver metastases that could lead to acute liver failure, we decided to add SSAs to chemotherapy, as first-line therapy.

Cytotoxic chemotherapy with 5-fluorouracil, capecitabine, dacarbazine, oxaliplatin, streptozocin and temozolomide can be used, in association with SSAs, in patients with progressive metastatic disease [26]. ESMO guidelines recommend CHT in advanced pancreatic NETs (pNETs) and in neuroendocrine neoplasms (NEN) G3 of any site [45]. A systematic review showed that patients treated with CHT for locally advanced or metastatic well-differentiated G1/G2 GI NETs experienced poor results (OR rate range 5.8–17.2%) [46]. The use of the CAPTEM regimen is associated with objective responses and promising survival outcomes in metastatic, well-differentiated, intermediate and high-grade NETs (two-year OS = 42%, median PFS = 10 months, median PFS = 17 months in patients who received CAPTEM in first line) [47]. More studies reported similar results regarding CAPTEM regimen in NEN, with significantly higher PFS, median OS, objective response rate and disease control rate in pNENs compared with non-pNENs [48,49]. Pancreatic NENs have a lower O (6)-methylguanine-DNA methyltransferase (MGMT) expression than non-pNENs, and this might be the rationale for a superior treatment response to CAPTEM regimen. Yet, the use of MGMT expression or promoter methylation as a sole predictor for response to CAPTEM is not recommended and more biomarkers are needed for clinical decision making [49,50].

CAPTEM regimen is rarely associated with serious toxicities (grade 3 or 4 stratified by Common Terminology Criteria for Adverse Events (CTCAE) v5.0 Grade) and has low discontinuation rates, even in patients who follow the treatment for more than a year [48]. The ENETs guidelines currently recommend this regimen for patients with intermediate and high-grade NETs as a second line therapy [51]. In a retrospective study, patients with metastatic intermediate and high-grade NETs (pancreatic and nonpancreatic included) treated with CAPTEM had median PFS of 15.9 months as compared to a median PFS of 3.1 months when subsequent lines of therapy were used [52]. Similarly, in a series of 30 patients with metastatic well or moderately differentiated pNETs who received CAPTEM in the first-line setting, the objective radiographic response rate was 70% with a median PFS of 18 months, a 2-year OS rate of 92% and a 12% rate of grade 3 and 4 toxicities [53]. This regimen clearly proves its superiority compared to other cytotoxic regimens (triplet combination of streptozocin, doxorubicin and 5-fluorouracil) with a response rate of 39%, a 2-year OS rate of 74% and a 23% rate of grade 3 and 4 toxicities [54].

RADIANT-4, a phase 3 study, found that treatment with everolimus is associated with significant improvement in PFS vs. placebo, in patients with advanced, progressive, well-differentiated, non-functional neuroendocrine tumors of lung or gastrointestinal origin [55]. The appropriate sequencing or combination of various drugs remains unclear and is mostly dependent on patient individual factors, such as comorbidities or side effects.

Ablative methods, including radiofrequency ablation and cryoablation, and hepatic artery chemoembolization with doxorubicin, cisplatin and mitomycin C, can result in tumor regression and control of the symptoms. Additionally, surgery is recommended for symptoms palliation [6]. PRRT is recommended for patients with inoperable or metastasized NETs. Lutetium-177 coupled with SSAs is a very promising approach in treatment of NETs [56]. The combined therapy with 177Lu-DOTATE plus octreotide LAR showed better tumor response rate vs. therapy with octreotide LAR alone [57]. Furthermore, combining PRRT with capecitabine and temozolomide (radiosensitizing chemotherapy) may be feasible with minimal incremental toxicity [58]. There are some inclusion criteria for PRRT, such as inoperable/metastatic, well differentiated (G1/G2) NET (well-differentiated G3 NET may be considered), sufficient tumor uptake on the diagnostic somatostatin scintigraphy, sufficient bone marrow reserves, creatinine clearance >50 mL/min, Karnofsky performance status >50, expected survival >3 months and signed informed consent [56]. In our case, due to no somatostatin analogue uptake on the scintigraphy, PRRT could not be considered.

## 4. Conclusions

In conclusion, the lesson to be learned from this case resides in the fact that somatostatinomas are rare NETs, seldom associated with somatostatinoma syndrome, which can sometimes occur in the settings of a genetic syndrome such as NF1. Despite the progress achieved in the management of metastatic, progressive, unresectable NETs, in general, and for somatostatinoma, in particular, prognosis is poor, and the optimal treatment of choice still remains a challenge for the multidisciplinary team involved.

## Figures and Tables

**Figure 1 diagnostics-10-00620-f001:**
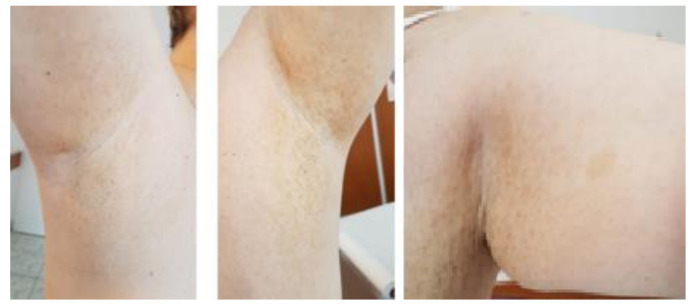
Bilateral axillary freckling and and café au lait spot on the left arm (scale 1:13).

**Figure 2 diagnostics-10-00620-f002:**
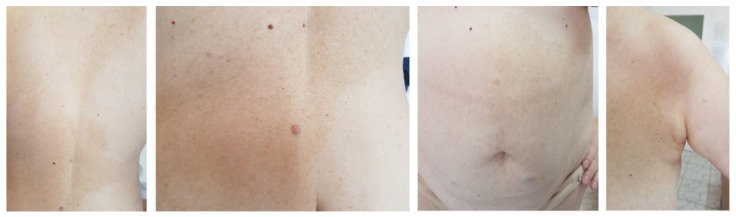
Large, poorly circumscribed areas of cutaneous hyperpigmentation and freckling on the trunk (scale 1:13/1:6/1:13/1:13).

**Figure 3 diagnostics-10-00620-f003:**
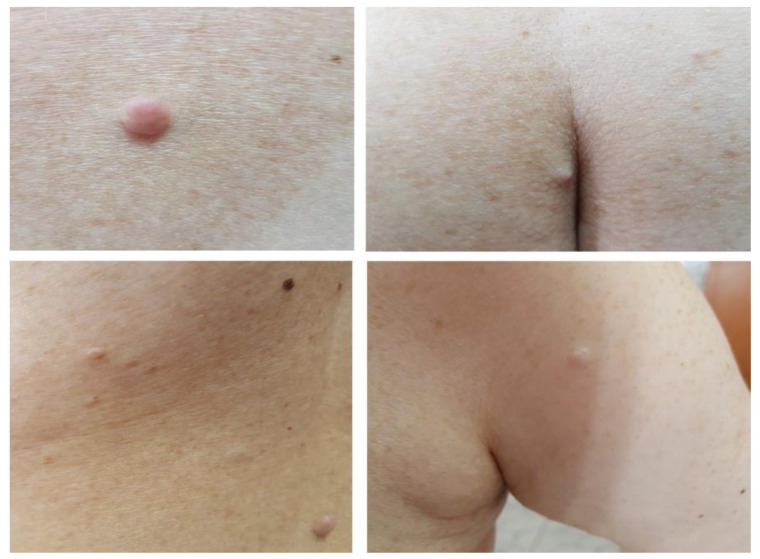
Multiple cutaneous neurofibromas distributed on the trunk and limbs (scale 1:7/1:13/1:11/1:11).

**Figure 4 diagnostics-10-00620-f004:**
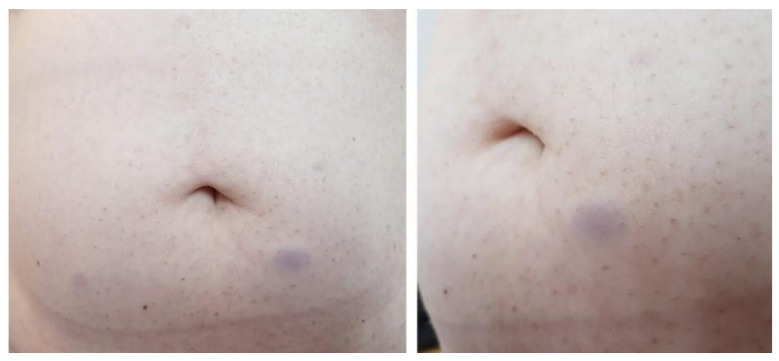
Asymptomatic, violaceous subcutaneous tumors in the abdominal area-pseudoatrophic macules typically found in NF1 (scale 1:11/1:8).

**Figure 5 diagnostics-10-00620-f005:**
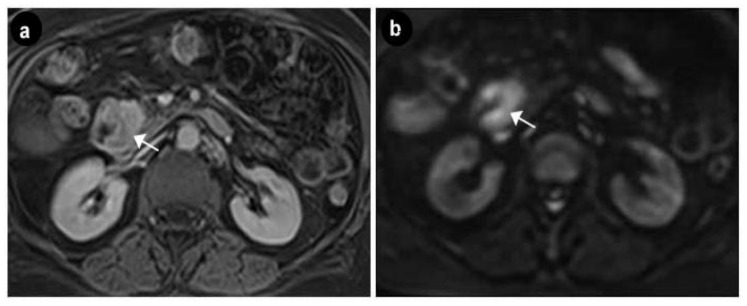
Axial MRI exam—duodenal tumor. (**a**) Gadolinium enhanced T1w image showing an hyperintense lesion in the arterial phase suggestive of the neuroendocrine etiology (arrow). (**b**) Diffusion Weighted Imaging (DWI) showing an hyperintense lesion with intense diffusion restriction (arrow).

**Figure 6 diagnostics-10-00620-f006:**
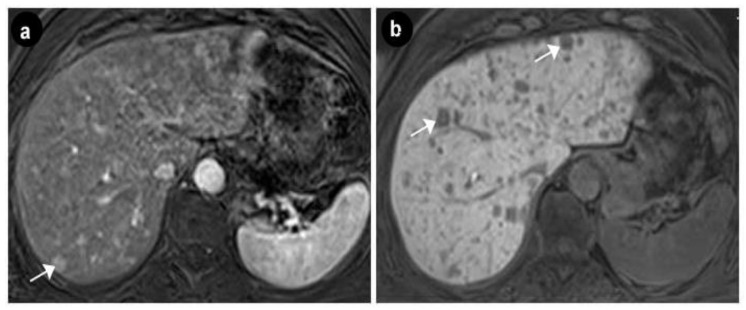
Axial MRI exam—multiple hepatic nodular lesions suggesting miliary metastic disease. (**a**) Gadolinium enhanced T1w showing hypervascular lesions (arrow). (**b**) T1w image, hepato-biliary phase after gadolinium enhancemet, showing hypointense lesions suggestive of the tumoral etiology (arrows).

**Figure 7 diagnostics-10-00620-f007:**
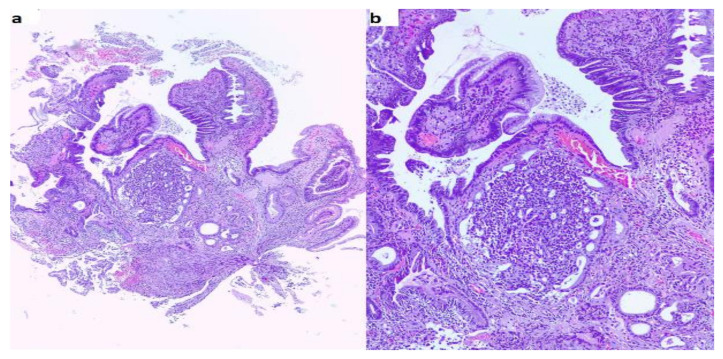
Histological findings. (**a**) HE, 40X. Duodenal biopsy with glandular and cribriform structures in the lamina propria, with the infiltrative aspect. (**b**) HE, 100X. Detail of the previous picture with cribriform structures and focal foveolar metaplasia of the villous epithelia.

**Figure 8 diagnostics-10-00620-f008:**
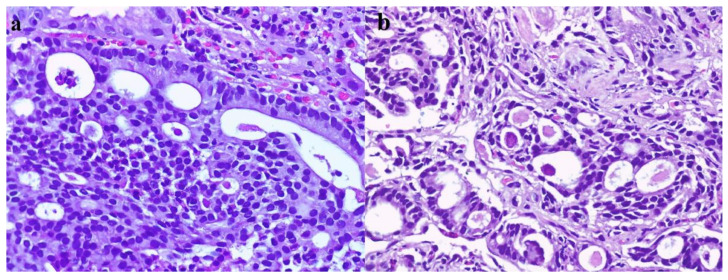
Histological findings. (**a**) HE, 200X. Cribriform structure with cuboidal or low columnar epithelial cells, with eosinophilic cytoplasm and round, central, monotonous nuclei, without visible nucleoli. Some necrotic cells and calcospherites are visible in the glandular lumina. (**b**) Glandular (acinar) growth pattern of the tumor with calcospherites (psammoma bodies).

**Figure 9 diagnostics-10-00620-f009:**
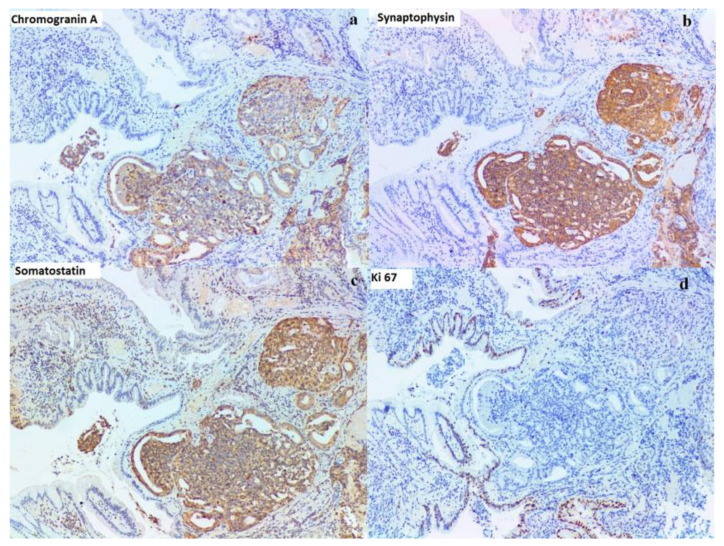
Imnuohistochemistry findings 40X. (**a**) Chromogranin A (Cell Marque, monoclonal, dilution 1:100): positive reaction of the tumoral cells, cytoplasmic, with internal positive control. (**b**) Synaptophysin (Immunopath, monoclonal, dilution 1:500): positive reaction of the tumoral cells, cytoplasmic, with internal positive control. (**c**) Somatostatin (Abcam, monoclonal, dilution 1:200): positive reaction of the tumoral cells, cytoplasmic, with internal positive control. (**d**) Ki67 (Leika, monoclonal, dilution 1:100): 1% in tumoral cells, with internal positive control in cryptic epithelial cells.

**Figure 10 diagnostics-10-00620-f010:**
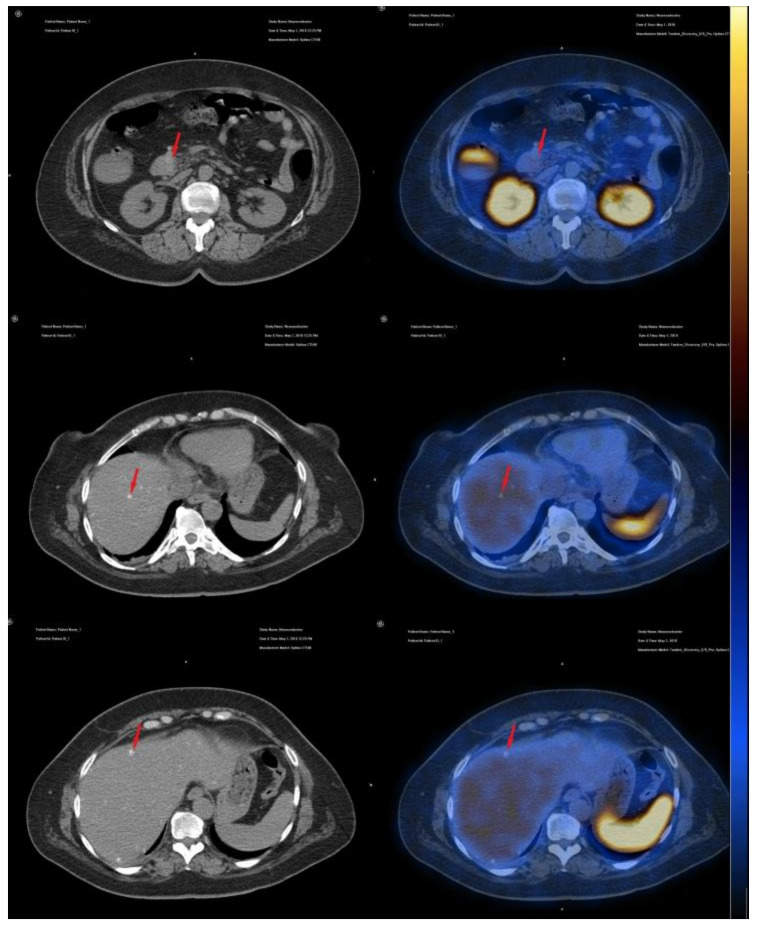
Axial unenhanced CT and fused SPECT-CT images showing duodenal tumor and multiple small hepatic metastasis with no uptake of 99mTc-EDDA/HYNIC-Tyr3-Octreotide (red arrows).

**Figure 11 diagnostics-10-00620-f011:**
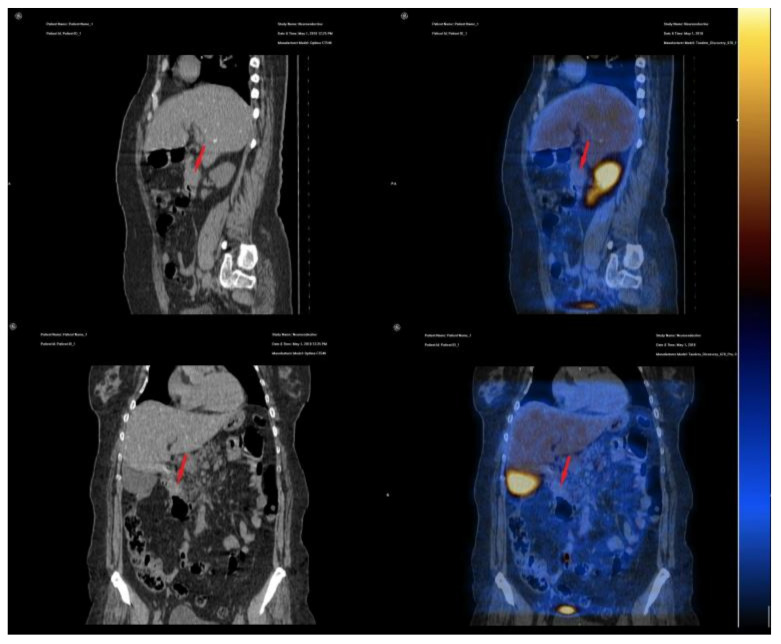
Sagital and coronal CT and SPECT-CT acquisition; duodenal tumor without 99mTc-EDDA/HYNIC-Tyr3-Octreotide uptake, suggesting no expression of SSTR2 or SSTR5 (red arrows).

**Figure 12 diagnostics-10-00620-f012:**
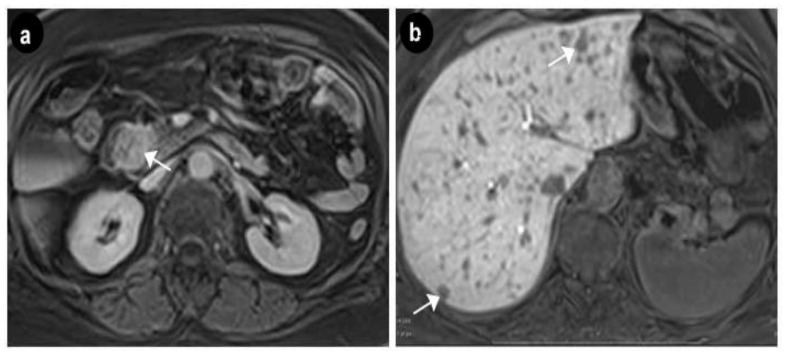
Follow up MRI at 3 months. (**a**) Slight regression of the duodenal tumor on T1w image after gadolinium enhancement (arrow). (**b**) Minimal regression of the metastatic hepatic lesions on the hepato-biliary phase (arrows).

**Figure 13 diagnostics-10-00620-f013:**
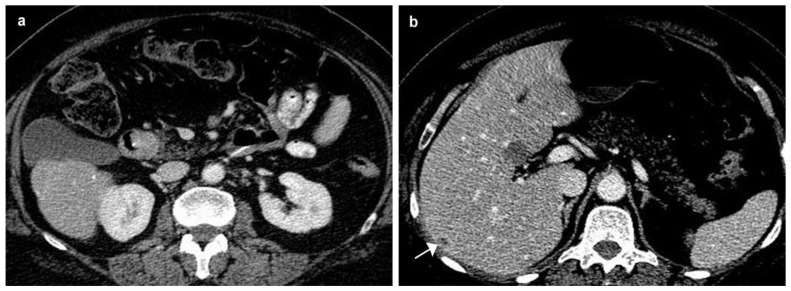
Follow up CT scan at 16 months from diagnosis (**a**) CECT - arterial phase- stable duodenal tumor. (**b**) CECT-portal phase-nodular lesion located in the sixth segment of the liver with an enhancing peripheral rim and a hypovascular center suggestive of a new metastasis (arrow).

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
