# Peer review of "Somatostatinoma and Neurofibromatosis Type 1-A Case Report and Review of the Literature"

_diagnostics, 2020, doi:10.3390/diagnostics10090620_

Round 1
Reviewer 1 Report
The authors report a adult patient with NF1 presenting with Somatostatinoma
General impressions,
The manuscript is well written and the clinical case well reported yet I believe it might result more readable if more synthetic in some parts (introduction, discussion). For example very long paragraphs on therapy, outcome when instead I would have expected to read more about the association with NF1.
In the conclusion section authors do not mention NF1, and I should reenforce e give the final message that facing such a rare tumor it is mandatory to exclude genetic predisposing disorder that might be not yet diagnosed as in their patient.
This is fundamental, because they give to the association NF1 and somatostinoma a predominant role in the title.
Moreover in the title they mention a review of literature but they do not discuss this point and do not cite all references after that by Tanaka et al , 2000
The blu macules reported are pseudoatrophic macules typically found in NF1 https://www.ncbi.nlm.nih.gov/pmc/articles/PMC2716546/
Minor revision
Line 22 tel 0040722714897; mirela_gherghe@yahoo.com; 0040720544679
Please put the abbreviation “tel” for dr gherghe too
For the title why you do not use: more than -?
“Somatostatinoma and Neurofibromatosis type 1: a case report and review of the literature
Abstract
-I would erase measurements of the nodule at CT.
-" if it is a solitary disease or is just on fragment of a genetic syndrome"- pheraps it would be more appropriate talk about sporadic lesion or just one feature/complication/manifestation of a predisposing genetic disorder
Line 55 alteration —> mutations
Line 56 “which determines the inactivation of p21 RAS “ actually neurofibromin negatively regulates Ras activity by catalyzing the hydrolysis of RAS-GTP , and thus down regulate the MAPK cascade.
At this point it would be nice to specify that NF1 can lead to several endocrine disorders, in infancy mainly due to tumors involving the hypothalamic regions (OPG) [J Clin Endocrinol Metab . 2020 Jun 1;105(6):dgaa138. doi: 10.1210/clinem/dgaa138. Pretreatment Endocrine Disorders Due to Optic Pathway Gliomas in Pediatric Neurofibromatosis Type 1: Multicenter Study] and in adult-age involving neuroendocrine system
Line 66 GIP and line 75 CT/ MRI, and line 83 PET
please check if the acronym has been extensively report before its first use
Line 87 the following sentence “Usually, at diagnosis, patients present with metastatic disease, but survival rate is high, especially in those with metastatic duodenal somatostatinoma “ should be referenced
Line 133 -138 I would avoid to list clinical criteria, just assessing that the clinical diagnosis was made according the still valid NIH diagnostic criteria
Line 151 and others, I suggest to use this kind of formula AxBxC for reporting measurement , for example “with maximum diameters of 12/10 mm “ “with maximum diameters of 12x10 mm”
Line 208 “equipped with LEHR (low energy high resolution) collimator “—>“equipped with low energy high resolution (LEHR) collimator “
Line 288 author should reference this statement
Even if it is clear for reader, for each figure specify what it represents , for ex
Regarding nf1 diagnosis, Please specify if patient did not undergo to molecular analysis to confirm the clinical diagnosis of NF1, iris nodules were searched?
In general I suggest to use NF1 more than NF-1, anyway authors should chose one between the two
Figures
In general report in the legend “arrow”, “red arrow” etc in parenthesis, given you correctly put them on figure
Figure 5 and. 6specify the they are axial images
Figure 6 abdominal MRI , figure 7 histological findings ….could you merge giure 8 and 9 if they both refer to tumor
Author Response
Thank you very much for the interest in our paper and for the valuable suggestions.
General impressions
We did our best to improve the introduction and the discussion sections as highlighted in text. We also updated and rearranged the references accordingly.
Line 418 – we mentioned NF1 in conclusions
Lines 129, 130, 142-143- Figure 4- we added the comment ‘pseudoatrophic macules typically found in NF1’
Minor revision
Line 23- we added tel for dr Gherghe
Line 31- we erased the measurements of the nodule at CT
Line 39-40- we changed it to ‘When a patient is diagnosed with a rare NET, such as a somatostatinoma, is of utmost importance to determine if it is a sporadic tumor or just a feature of a genetic disorder.’
Line 58- we modified it to ‘it is caused by mutations in the NF1 gene’.
Lines 58- 64 we rearranged the paragraph to ‘. NF1 is localized on chromosome band 17q11.2 and it acts as a tumour suppressor gene. Biallelic NF1 inactivation results in complete loss of functional neurofibromin activity. Neurofibromin acts as a negative regulator of the RAS–MAPK pathway and it has a GTPase activating domain that interacts with RAS and converts active RAS–GTP to its inactive form. Apart from NF1 promotor hypermethylation, other somatic inactivation events occur in tumours associated with NF1, such as intragenic mutations and loss of heterozygosity.’
Line 64-67 we added ‘Persons with NF1 develop both benign and malignant tumors at increased frequency throughout life, neurofibromas being the most common type of benign tumor. Optic pathway gliomas are the predominant type of intracranial neoplasms and it can lead to endocrine disorders mostly in pediatric population’ and the corresponding reference Santoro, C; Perrotta, S; Picariello, S; et al. Pretreatment Endocrine Disorders Due to Optic Pathway Gliomas in Pediatric Neurofibromatosis Type 1: Multicenter Study. J Clin Endocrinol Metab. 2020;105(6):dgaa138. doi:10.1210/clinem/dgaa138
Line 76, 85, 86, 90, 93 we checked and modified the acronyms
Line 96 we referenced “Usually, at diagnosis, patients present with metastatic disease, but survival rate is high, especially in those with metastatic duodenal somatostatinoma “
Lines 145- 148 we changed it to ‘The mucous membranes did not show any pathologic changes and Lisch nodules were absent. Based upon the presence of 3 out of 7 characteristic clinical features our patient met diagnostic criteria developed by the United States National Institutes of Health (NIH) Consensus Conference in 1987 and updated in 1997 and was clinically diagnosed with neurofibromatosis type 1 (NF1).’
Lines 155, 161 we changed the dimensions reporting
Line 206-208 we modified it to ‘..low energy high resolution (LEHR) collimator and SPECT-CT acquisition 2 hours after injection, on the region of interest (abdomen). On somatostatin receptor scintigraphy (SRS)..’
Line 332– we referenced the statement ‘Our patient had normal plasma normetanephrines and metanephrines, but further regular screening for other malignancies associated with NF1 is mandatory’ Stewart, D.R.; Korf, B.R.; Nathanson, K.L.; Stevenson, D.A.; Yoha,y K. Care of adults with neurofibromatosis type 1: a clinical practice resource of the American College of Medical Genetics and Genomics (ACMG). Genet Med. 2018;20(7):671-682. doi:10.1038/gim.2018.28
268-277 we introduced the paragraph ‘The multidisciplinary team management of this particular case also included genetic counseling. Genetic testing can be considered for a molecular diagnosis. Genetic testing is often not required but can be useful in confirming the diagnosis for children who do not meet diagnostic criteria or only demonstrate café-au-lait macules and axillary freckling. Genetic testing can be performed to confirm the diagnosis in questionable cases and to help direct screening of family members. Although of great academic interest, given the unquestionable clinical diagnosis, the fact that our patient did not have any first degree relatives that could benefit and that a positive NF1 mutation test does not predict the severity or complications of the disorder, with some specific exceptions, we decided not to pursue a genetic testing.’
Line 145- ‘The mucous membranes did not show any pathologic changes and Lisch nodules were absent.’
We used NF1.
Figures
We added “arrow” and “red arrow” in the legends.
We specified axial images when appropriate.
We modified as suggested the legend for Figure 5, 6, 7, 8, 9. We merged figure 8 and 9.
Reviewer 2 Report
This is a useful article and deserving publication. Nevertheless some revision is required.
English language and style are in general fine, but minor spell check is required.
For example: In abstract on line 27 it would be better to write: significant weight loss rather than: important weight loss.
In the case presentation section, only the patient's symptoms and test results should be reported.
The information contained in lines 133-138 - does not specify which of the symptoms occurred in the presented patient, but rather suggests that this is a presentation of the diagnostic criteria for NF1. This information should be included in the discussion section. Similarly, lines 163-168 and 199-204.
Moreover, it has not been said whether genetic testing for MEN1 has been performed.
Coexistence of NF1 with multiple synchronous and/or metachronous tumors: NETs, GIST, other benign or malignant neoplasms has been reported in recent years. It would be worth mentioning these papers in the discussion. The diagnostics and treatment patients with multiple neoplasms is still a great challenge for contemporary medicine. So this issue requires further research.
Author Response
Thank you very much for the interest in our paper and for the valuable suggestions.
Line 29 We changed ‘important’ with ‘significant weight loss’
In the case presentation section we kept only the patient's symptoms, clinical features and test results and we moved the information contained in lines 133-138, 163-168 and 199-204 in the discussion section lines 260-266, 319-324 and lines 330-336.
Line 222 we mentioned that ‘we performed an endocrinological screening for MEN1 and pheochromocytoma which were negative’. In this context, genetic testing for MEN1 was not recommended or performed.
Line 276 - Considering your recommendation, the coexistence of NF1 with multiple synchronous and/or metachronous tumors and the permanent need for care in these patients we cited a recent paper on this topic: Stewart DR, Korf BR, Nathanson KL, Stevenson DA, Yohay K. Care of adults with neurofibromatosis type 1: a clinical practice resource of the American College of Medical Genetics and Genomics (ACMG). Genet Med. 2018;20(7):671-682. doi:10.1038/gim.2018.28
We rearranged all the references.
Reviewer 3 Report
Thank you for this very intresting and original case report, I just would like to suggest to put those 2parts in the discussion rather than in the case presentation :
“Duodenal neuroendocrine tumors account for about 20% of gastrointestinal tract neuroendocrine tumors and may show G-cell differentiation (gastrinomas) or D-cell differentiation expressing somatostatin, termed somatostatinomas if theyare functional. Acinar and psammomatous somatostatinomas are commonly found in the periampullary region in patients with NF1 and represent one third of somatostatin expressing tumors. Sometimes patients with NF1 may present other mesenchymal lesions: neurofibromas and gastrointestinal stromal tumors. “
“Somatostatin receptor scintigraphy is a molecular imaging procedure for NET diagnostic and staging, more sensitive and specific at the biological than anatomical level, in contrast to conventional imaging, which it complements; the detection rate was reported to be between 80-100% in different studies [21-23]. Thereare two radiopharmaceuticals available on the market 111In-pentetreotide ([111In-DTPA0]-octreo-tide) with affinity for SSTR 2 and SSTR 5 and 99mTc-EDDA/HYNIC-Tyr3-Octreotide, with high affinity for SSTR 2, but with a lesser extent for203SSTR 3 and SSTR”
And to modify the writting of octreotide:"202 111In-pentetreotide ([111In-DTPA0]-octreo- tide)"
Author Response
Thank you very much for the interest in our paper and for the valuable suggestions.
We moved the information contained in lines 133-138, 163-168 and 199-204 in the discussion section lines 260-266, 319-324 and lines 330-336
We modified the line 334 into '111In-pentetreotide ([111In-DTPA0]-octreotide)'.
We rearranged all the references.